# Effects of Dietary Iron Level on Growth Performance, Immune Organ Indices and Meat Quality in Chinese Yellow Broilers

**DOI:** 10.3390/ani10040670

**Published:** 2020-04-12

**Authors:** Xiajing Lin, Zhongyong Gou, Yibing Wang, Long Li, Qiuli Fan, Fayuan Ding, Chuntian Zheng, Shouqun Jiang

**Affiliations:** Institute of Animal Science, Guangdong Academy of Agricultural Sciences, State Key Laboratory of Livestock and Poultry Breeding, Key Laboratory of Animal Nutrition and Feed Science in South China, Ministry of Agriculture and Rural Affairs, Guangdong Public Laboratory of Animal Breeding and Nutrition, Guangdong Key Laboratory of Animal Breeding and Nutrition, Guangzhou 510640, China; linxiajing728@sohu.com (X.L.); yozhgo917@163.com (Z.G.); wangyibing001@gmail.com (Y.W.); leeloong1985@sina.com (L.L.); fanqiuli_829@163.com (Q.F.); dingfa@sina.com (F.D.); zhengcht@163.com (C.Z.)

**Keywords:** Chinese yellow broiler, iron, growth performance, immune organ, meat quality

## Abstract

**Simple Summary:**

Iron (Fe) is an essential trace mineral for all living organisms, playing important roles in oxygen and electron transport as well as in DNA synthesis. At present, requirements of chickens for trace elements are still based on past National Research Council (NRC) (1994) standards. It is important, therefore, to accurately re-evaluate and satisfy the Fe requirements of modern broiler chicken, and especially important, non-Western breeds such as Chinese yellow broilers, for their optimal growth and health. Therefore, the aim of the present study was to investigate the effects of Fe on growth performance, immune organs and meat quality in Chinese yellow broilers. Based on the findings, feeding yellow-feathered broilers on a conventional corn-soy-based diet satisfies their requirements without additional Fe at ages 1 to 21, and 22 to 42 d, while 90 mg/kg in the finisher phase improved meat quality, and from the QP (quadratic polynomial) models of the key meat quality variables pH of breast muscle and drip loss of breast muscle, the optimal dietary Fe level was 89 to 108 mg/kg, and daily Fe fed allowance was 11 to 13 mg in the finisher phase (43 to 63 d).

**Abstract:**

The objective of three trials was to investigate the effects of dietary Fe on growth performance, immune organ indices and meat quality of Chinese yellow broilers during the whole growth period. A total of 1440 1-day-old, 1440 22-day-old, and 1080 43-day-old Lingnan yellow male broilers were randomly assigned to one of six dietary treatments with six replicates per treatment (40 birds per replicate for both 1 to 21 d and 22 to 42 d, 30 birds for 43 to 63 d). Additional Fe (0, 20, 40, 60, 80, and 100 mg/kg) was added as FeSO_4_ • H_2_O to the three basal diets (calculated Fe 50 mg/kg, analyzed 48.3, 49.1, 48.7 mg/kg, respectively). The calculated final dietary Fe concentrations in Starter, Grower and Finisher phases were 50, 70, 90, 110, 130, and 150 mg/kg. The results showed that average daily gain (ADG), average daily feed intake (ADFI) and feed conversion rate (FCR) of the broilers were not influenced by the different levels of Fe (*p*> 0.05). Weight indices of the spleen, thymus and bursa of Fabricius were not influenced (*p* > 0.05) by the different levels of Fe during three 21-day experimental periods. Hematocrit, and Fe contents of the liver and kidney were not affected by different levels of Fe (*p*> 0.05). The diet with 150 mg/kg of Fe increased the a* (relative redness) value of breast muscle compared to the 50 and 70 mg/kg diets at 24 h post mortem (*p*< 0.05). The diet with 90 mg/kg Fe increased the pH of breast muscle compared to broilers fed 50 or 150 mg/kg Fe (*p* < 0.05) 45 min after slaughter. The diet with 90 mg/kg Fe decreased drip loss of breast muscle compared to 150 mg/kg Fe (*p*< 0.05). These data suggest that feeding yellow-feathered broilers on a conventional corn-soy based diet satisfies their requirements without additional Fe at ages 1 to 21, and 22 to 42 d, while 90 mg/kg in the finisher phase improved meat quality, and from the QP (quadratic polynomial) models of the key meat quality variables, pH of breast muscle and drip loss of breast muscle, the optimal dietary Fe level was 89 to 108 mg/kg, and daily Fe fed allowance was 11 to 13 mg in the finisher phase (43 to 63 d).

## 1. Introduction

Iron (Fe) is an essential trace mineral for all living organisms, playing important roles in oxygen and electron transport as well as in DNA synthesis [1]. When poultry are fed Fe-deficient diets or it is poorly absorbed, physiological function is degraded, Fe deficiency anemia and other diseases are more likely, with serious impacts on the production of animals [2]. At present, requirements of chickens for trace elements are still based on past National Research Council (NRC) (1994) [3] standards. It is important, therefore, to accurately re-evaluate and satisfy the Fe requirements of modern broiler chickens, and especially so for non-Western breeds such as Chinese yellow broilers, for their optimal growth and health.

Limited, and now old, research data focus on Fe nutrition of poultry, and available recommendations on dietary Fe for broilers vary widely, from 45 to 136 mg/kg [4,5,6,7,8]. Current requirements for Fe target fast-growing broilers such as Arbor Acres, Ross, and Cobb (42 d marketing age, average daily gain (ADG): 60 g, average daily feed intake (ADFI): 95 g, feed conversion rate (FCR): 1.6). Corresponding information on dietary Fe for slower-growing Chinese Yellow broilers (63 d marketing age, average daily gain (ADG): 36 g, average daily feed intake (ADFI): 80 g, feed conversion rate (FCR): 2.2) is lacking, despite China being the third largest producer of chickens in the world; Chinese yellow-feathered broilers are a very important meat resource, with more than 4 billion being produced per year in China.

Fe, a vital component of hemoglobin in erythrocytes, is required for transporting oxygen around the body [9,10], using both hemoglobin and myoglobin [11,12] for the delivery, storage and use of oxygen in muscles [13]. Hemoglobin and myoglobin play an important role in maintaining normal meat color, the most visual indicator of meat quality [14]. There are few studies on the effects of dietary Fe on meat quality, especially in broiler chickens. Apple et al. [15] found that increasing the level of Fe in the diet of growing pigs significantly increases the a* value of meat color. High doses of Fe in the diet of swine significantly increase the level of fat peroxidation in fresh and cooked meat [16]. The present study, therefore, sought to investigate the effects of Fe on growth performance, immune organs and meat quality in Chinese yellow broilers.

## 2. Materials and Methods 

### 2.1. Ethics Approval

For the experimental field on animals, all protocols for all animal experiments were approved by the Scientific Ethics Committee of the Institute of Animal Science, Guangdong Academy of Agricultural Sciences, Guangdong Province, China (GAASISA-2015-03), and the laboratory experiments and protocols were handled in accordance with the guidelines established by the Institute of Animal Science, Guangdong Academy of Agricultural Sciences, Guangdong Province, China.

### 2.2. Chickens and Husbandry 

Three linked experiments were conducted to determine the effects of Fe level on growth performance, immune organ indices and meat quality during three growth stages of Chinese yellow male broilers (Lingnan, a genetically improved Chinese meat-type chicken, obtained from the Institute of Animal Science, Guangdong Academy of Agricultural Sciences, Guangzhou, China). Experiment 1 examined the starter phase (d 1 to 21, initial BW 40.4 ± 0.1 g); Experiment 2 covered the grower phase (d 22 to 42, initial BW 374 ± 2.67 g); and Experiment 3 involved the finisher phase for this breed (d 43 to 63, initial BW 1.21 ± 0.05 kg). Each experiment was conducted with a single factorial arrangement of six treatments, each consisting of six replicate floor pens (length 3.5 m × width 1.3 m), with 40, 40, and 30 birds (Experiments 1, 2, and 3 with total numbers 1440, 1440, and 1080). The litter consisted of wood shavings. The temperature was maintained at 32 °C during Experiment 1 (infrared warming light to keep the chicks warm) and was ambient in the other two experiments with older chickens. The birds were provided with ad libitum access to water and mashed diets throughout each 21-d experimental period.

### 2.3. Diets

The basal corn-soy protein concentrate meal mash diet for each experiment (Table 1) was based on the recommendations of Chinese Feeding Standard of Chicken for the three growth phases and was the feed for Treatment 1 (controls, calculated Fe 50 mg/kg, analyzed contents 48.3, 49.1, 48.7 mg/kg) in each experiment. Treatments 2 to 6 consisted of the basal diet supplemented with 20, 40, 60, 80, and 100 mg/kg of added FeSO_4_ • H_2_O (Guangdong Newland Feed Science & Technology Co., Ltd., Guangzhou, China); the six dietary Fe contents were assayed by atomic absorption spectrophotometry (Z-2000, HITACHI, Tokyo, Japan) after the diets were assembled. The drinking water was filtered over silica sand and then deionized to eliminate possible sources of Fe (The content of Fe in drinking water before treatment was 5 mg/L, after treatment was < 0.3 mg/L). The apparatus for removing Fe from water was from Guangzhou Chenxing Environmental Protection Technology Co., Ltd. (Guangzhou, China). Before the chickens entered Experiments 2 or 3, they had been raised on the same basal diets, which contained 80 mg/kg Fe.

### 2.4. Measurements

#### 2.4.1. Growth Performance

Birds were weighed at the beginning (d 1, d 22, and d 43) and end of each 21-d experimental period. Average daily feed intake (ADFI) was measured on a per pen basis from the 21-d feed consumed and the feed conversion ratio (FCR) was calculated. Mortality was checked daily and dead birds were weighed in order to adjust the calculated FCR.

#### 2.4.2. Sampling

After 16-h feed withdrawal, two chickens in Experiment 1, 2 and 3 were chosen at random (excluding obvious outliers in BW) from each pen, individually weighed and 7 mL blood was sampled from the brachial vein into evacuated tubes containing EDTA-K2 (1 mg/mL blood). One mL of non-clotted blood was held to measure hematocrit using a precision ESR (erythrocyte sedimentation rate) tube (inside diameter, 5 mm; length, 112 mm).

The birds were electrically stunned and exsanguinated to obtain tissues. The spleen, thymus and bursa of Fabricius were collected, rinsed with physiological saline solution, then weights were recorded after blotting with filter papers. In Experiment 3, when birds were of market size, liver and kidney were collected for measurement of Fe content, and breast muscle was collected to assess meat quality.

#### 2.4.3. Meat Quality

The pH of the right pectoralis major was measured by insertion of a needle probe (HI8424, Beijing Hanna Instruments Science & Technology Co., Ltd., Beijing, China). The drip loss of breast muscle was estimated 45-min postmortem as follows: a meat sample (about 1 g) was weighed, placed in sealed, air-filled plastic bags, held at 4 to 6 °C for 24 h, and drip loss was measured by re-weighing.

The color measurements of breast muscles were carried out 45 min after slaughter to measure CIE lab values (L* measures relative lightness, a* relative redness, and b* relative yellowness) using a Chroma Meter (CR-410, Minolta Co., Ltd., Osaka, Japan). Each sample chosen was measured at three locations and the values were averaged. 

The whole left breast muscle was collected, packed in plastic bags, refrigerated overnight at 4 °C and then brought to room temperature before cooking. The breast muscle from each bird was cooked to an internal temperature of 70 °C on a digital thermostat water bath (HH-4, Jiangbo instrument, Jiangsu, China). Endpoint internal temperature was monitored with a thermometer. Cooked muscle was cooled to room temperature. Slices of 1 × 1 cm were cut perpendicular to the fiber orientation of the muscle. Ten 1 × 1 cm segments about 3 cm thick were removed parallel to the fiber orientation through the thickest portion of the cooked muscle. Warner–Bratzler shear force was determined by using an Instron Universal Mechanical Machine (Instron model 4411, Instron corp., Canton, MA). This was attached to a 50 kg load cell and tests were performed at a cross head speed of 127 mm/min. Signals were processed with the Instron Series 9th software package.

#### 2.4.4. Biochemical Indices in Plasma, Liver and Kidney

The Fe content of liver and kidney were analyzed by colorimetric methods using assay kits (Nanjing Jiancheng Institute of Bioengineering, Nanjing, China), and the instructions provided. All samples were measured in triplicate, at appropriate dilutions.

### 2.5. Statistical Analysis

Replicate (or an average of the two sampled birds) was used as the experimental unit. The effect of treatment was examined by one-way ANOVA and, where appropriate, was followed by Tukey post-hoc tests (SPSS software version 17.0.1. Chicago, IL, USA). Means were considered to be significantly different at *p* < 0.05. All data were tabulated as means with SE derived from each ANOVA mean square error for n = 6. Based on the key indices of meat quality (pH of breast muscle, drip loss of breast muscle), quadratic polynomials (QP) were used to determine the optimal dietary content of Fe for chickens. The QP model (Y = α + β × Fe + γ × (Fe)^2^) had Y as the dependent variable; α was the intercept; β was the linear coefficient; γ was the quadratic coefficient. The response for Fe was defined as Fe = – β / (2 × γ).

## 3. Results

### 3.1. Growth Performance

The results for effects of Fe on performance of Chinese yellow broilers are presented in Table 2. ADG, ADFI and FCR of the broilers were not influenced (*p* > 0.05) by the different levels of Fe. 

### 3.2. Indices of Immune Organs 

Weight indices of the spleen, thymus and bursa of Fabricius of the broilers were not influenced (*p* > 0.05) by the different levels of Fe during three 21-d experimental periods (Table 3).

### 3.3. Biochemical Analyses

Table 4 shows that hematocrit, and Fe contents of liver and kidney in broilers were not affected by different levels of Fe (*p* > 0.05).

### 3.4. Meat Quality

Relevant indices of meat quality, viz. breast muscle, leg muscle, meat color, pH, drip loss and shear force are presented in Table 5. The diet with 150 mg/kg Fe increased a* value of breast muscle 24 h after slaughter compared to the 50 and 70 mg/kg Fe (*p* < 0.05). The diet with 90 mg/kg Fe increased the pH of breast muscle compared to broilers fed 50 or 150 mg/kg Fe (*p* < 0.05) at 45 min postmortem. The diet with 90 mg/kg Fe decreased the drip loss of breast muscle compared to broilers fed 150 mg/kg Fe (*p* < 0.05). There were no significant effects on other indicators of meat quality at 45 min and 24 h after slaughter (*p* > 0.05).

### 3.5. Regression Analyses 

The data for the pH of breast muscle firstly increased and then decreased with the increase of the level of dietary Fe. Conversely, the data for drip loss decreased first and then increased. The data change for pH and drip loss of breast muscle accorded with the QP models, therefore, they were selected out for further analysis by QP regressions related to the dietary Fe level (Figure 1 and Figure 2, Table 6). According to the optimal dietary Fe response from regression models in Table 6 and the average daily feed allowance of 120 g, the daily Fe fed allowance of Chinese Yellow broilers were 12.96 or 10.68 mg during 43 to 63 days. 

## 4. Discussion

The effect of dietary Fe level on animal growth performance has been inconsistent in previous studies. Farrow et al. [17] reported that no significant effect on growth performance with 2500 mg/kg supplementary Fe in White Rock chickens. Cao et al. [18] found that the addition of high doses (400, 600 and 800 mg/kg) of Fe in the control diet (Fe 188 mg/kg) significantly increased BW and ADFI in Ross broilers. In contrast, Gou et al. [19] showed that corn-soybean diets (Fe 245 to 1651 mg/kg) had no effect on ADFI, ADG and F:G of Lingnan broilers from d 1 to d 21 of age indicating that different breeds of chickens had different tolerance and requirements for Fe. Ma [20] reported that the level of dietary Fe (47–147 mg/kg) had no effect on the growth performance of AA broilers at 22 to d 42 of age with that broilers were fed the same diet (containing 127 mg/kg Fe) from 1 to d 21 of age. Sun et al. [21] found that the level of dietary Fe (150–250 mg/kg) significantly increased ADG and FCR in AA broilers at 22 to d 42 of age with that broilers were fed the same diet (containing 186 mg/kg Fe) from 8 to d 21 of age, these studies indicated that the retention of iron in broilers from previous periods had no obvious effect to assess Fe requirements in growth performance. The results of the present experiment showed that different levels of dietary Fe (50 to 150 mg/kg) had no significant impact on the growth performance of yellow broiler chickens during 1 to 21, 22 to 42, or 43 to 63 d, indicating that the Fe content in the basic diet was sufficient for the growth of these chickens. The results of the present experiment showed that different levels of dietary Fe (50 to 150 mg/kg) had no significant impact on the growth performance of yellow broiler chickens during 1 to 21, 22 to 42, or 43 to 63 d, indicating that the Fe content in the basic diet was sufficient for the growth of these chickens. The present study used a corn-based diet, where the Fe content of corn was 32 to 36 mg/kg, whereas other energy sources had higher Fe contents, sorghum 45 to 59 mg/kg and barley 48 to 100 mg/kg [22,23]. It is recommended, therefore, that no supplemental Fe is needed for the basal diet used here for yellow-feathered broiler chickens over their entire growth to marketing at 63 d.

The immune organs of broilers include the thymus, spleen and bursa of Fabricius. The thymus is the central immune organ of broilers producing T cells and mononuclear macrophages [24]. The spleen is the largest peripheral immune organ in poultry and contains abundant lymphocytes [25]. The size of the spleen can reflect the immune status of the body [26]. It is generally believed that a large immune organ index indicates a well-developed organ with strong immune function [27,28]. There is little known of the influence of Fe on immune organs in animals, particularly in broiler chickens. Fe deficiency does affect the development of immune organs and reduces mammalian resistance to diseases [29]. Fe excess can inhibit the bactericidal activity of white blood cells, reduce the bacteriostatic effect of ferritin and lactoferrin, and increase the incidence and severity of infection [30]. Feng et al. [31] reported that 60, 90, 120 mg/kg of Fe-Gly in piglet diets significantly increases thymic relative weight compared with the basal diet without Fe addition. Rothenbacher and Sherman [32] found that Fe could accelerate the development of immune organs and Fe deficiency tended to cause thymic atrophy in rats. The thymus, bursa of Fabricius and spleen of Avian chicks atrophied with Fe-deficient diets [33]. The weight of the thymus was affected by increasing dietary Fe-Gly levels (the range was 40 to 160 mg) but the spleen index linearly decreased with increasing dietary Fe-Gly [34]. The present study showed no effect of dietary Fe content (50 to 150 mg/kg) on the immune organs of yellow-feathered broiler chickens; the Fe content in this un-supplemented conventional feed diet meets requirements for immune development and likely function.

A large number of studies have shown that hematocrit is a sensitive index of the Fe nutritional status [35,36,37,38]. Ma [5] reported that the hematocrit significantly increases with the dietary Fe content of Arbor Acres broilers (5 to 45 mg/kg) from d 1 to d 14 of age. Ma [20] showed that there was no significant effect on the hematocrit of AA broiler chickens with a corn-soybean meal diet (40 to 140 mg/kg) from d 22 to d 42 of age. These studies suggest that Fe has a significant effect on the hematocrit only when the body is extremely deficient in Fe. The present experiment similarly showed that Fe (50 to 150 mg/kg) had no effect on hematocrit of yellow-feathered broiler chickens aged 1 to 63 days; additionally, there was no effect on Fe content of the kidney and liver, sensitive and reliable tissues reflecting Fe deposition [39,40]. Ma [5] found that Fe content in the kidney of AA broiler chickens aged 22 to 42 d was not significantly affected by dietary Fe level (40 to 140 mg/kg). Seo et al. [41] found that a high dose of ferrous methionine significantly increased hepatic Fe content in Ross broiler chickens; Fumgouri et al. [42] came to a similar conclusion. However, Ma [5] reported that the Fe content in liver of AA broilers gradually decreased with increased dietary Fe, above 120 mg/kg, possibly because the liver could maintain the balance of Fe, avoiding excessive Fe deposition and causing body damage. The inconsistency of the preceding studies may reflect the variety and feed type of the broilers.

At present, little is known of the effect of Fe on meat quality. pH is a key indicator of meat quality, having a great influence on the quality of chicken carcasses [43,44]. Water holding capacity and meat color are both influenced by pH [45]. Within a certain range, higher pH of meat results in reduced drip loss. An important biochemical change in the muscle of broilers is the decrease of pH postmortem, reflecting glycolytic rate of the muscle. Fe is essential for oxygen transport and is a key component of superoxide dismutase (SOD), which has an important influence on the oxidation of the body [46]. In the present experiment, the lowest drip loss and highest pH occurred with 90 mg/kg Fe, and differences existed with 50 and 150 mg/kg Fe, indicating that there was an optimal level of dietary Fe, in terms of drip loss and pH, and therefore meat quality.

Meat color is affected by myoglobin content and oxidation state [47] because iron, a key component of hemoglobin and myoglobin, underlies meat color. Wensing et al. [48] found that adding 50 mg/kg of Fe throughout the finishing phase could lead to over-red beef at slaughter. Apple et al. [15] fed finishing pigs a diet with amino acid-chelated Fe (50, 100, 150 mg/kg) with no effect on a* value during the shelf life. Seo et al. [40] showed that the addition of 100 or 200 mg/kg of ferrous methionine in the diet of Ross broilers reduced the L* value of the breast and leg muscles and increased the a* and b* values, but without significant difference with the controls. The present experiment also indicated that increasing dietary Fe significantly increased the a* value of the breast muscle, consistent with previous reports, demonstrating that Fe has an obvious effect on the redness of the breast muscle in yellow-feathered broilers.

## 5. Conclusions

Dietary addition of different levels of Fe (50 to 150 mg/kg) caused no significant difference in the growth performance, immune organ indices, hematocrit, or Fe content in liver and kidney of yellow-feathered broiler chickens. During the finishing phase (43 to 63 d), an appropriate amount of dietary Fe (90 mg/kg) improved indices of meat quality of breast muscle in these chickens. Overall, the results obtained here demonstrate that yellow-feathered broilers fed conventional diets based on corn and soy meal (Fe 50 mg/kg) require no additional Fe during starter (1 to 21 d) and grower (22 to 42 d) phases, while 90 mg/kg in the finisher phase improved meat quality, and from the QP models of the key meat quality variables pH and drip loss of breast muscle, the optimal dietary Fe level was 89 to 108 mg/kg, and daily Fe fed allowance was 11 to 13mg in the finisher phase (43 to 63 d).

## Figures and Tables

**Figure 1 animals-10-00670-f001:**
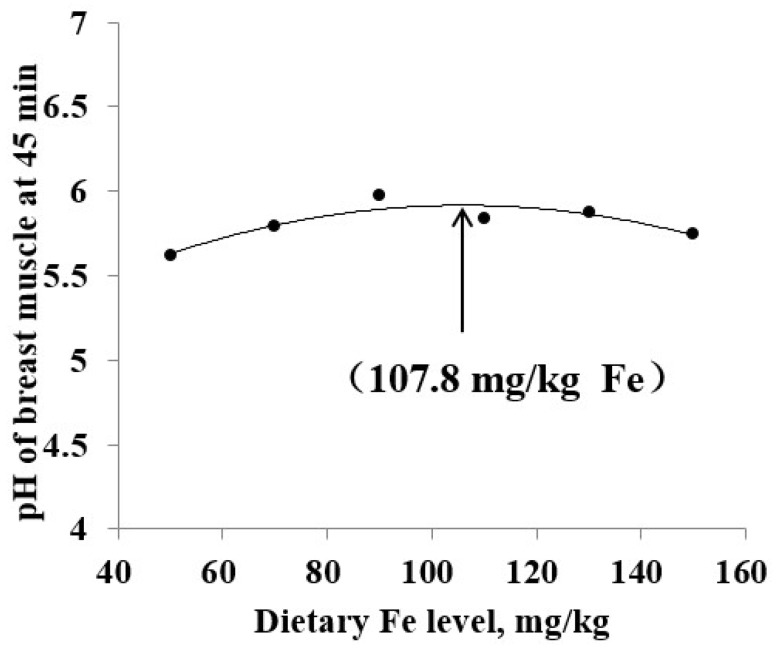
pH of breast muscle (at 45 min after slaughter) of broiler chicken fed diets supplemented with Fe. Regression equations obtained using the increasing dietary Fe in the current study (50, 70, 90, 110, 130 and 150 mg/kg). The quadratic polynomial (QP) regression (Y = 4.8897 + 0.0194 × X – 0.0009 × X^2^; the maximum response arrow pointing at 107.8 mg/kg Fe, R^2^ = 0.831, *p* = 0.064).

**Figure 2 animals-10-00670-f002:**
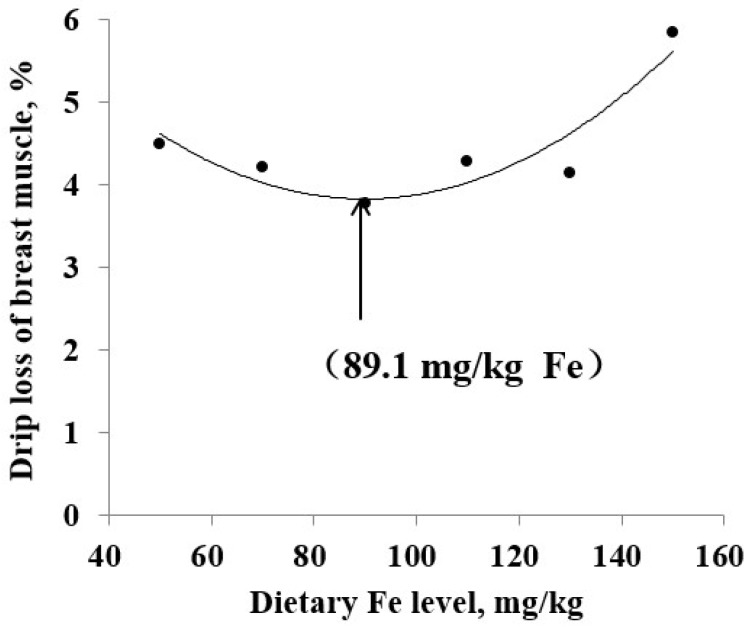
Drip loss of breast muscle (%) of broiler chicken fed diets supplemented with Fe. Regression equations obtained using the increasing dietary Fe in the current study (50, 70, 90, 110, 130 and 150 mg/kg). The quadratic polynomial (QP) regression (Y = 7.8401 – 0.0891 × X + 0.0005 × X^2^; the minimum response arrow pointing at 89.1 mg/kg Fe, R^2^ = 0.841, *p* = 0.069).

**Table 1 animals-10-00670-t001:** Composition and nutrient level of the basal diet for three experiments (as fed-basis).

Phases	Starter	Grower	Finisher
Ingredients, %	1 to 21 d	22 to 42 d	43 to 63 d
Corn	61.50	63.30	67.05
Soy protein concentrate (CP65%)	9.4	9.4	9.4
Feather meal	8.0	5.5	2.0
DDGS	15.7	15.7	15.0
Soybean oil	0.40	1.20	1.71
L-lysine HCL	0.29	0.26	0.19
L-threonine	0.28	0.36	0.43
DL-methioine	0.15	0.10	0.06
Calcium carbonate	1.15	1.00	0.93
Dicalcium phosphate	1.82	1.70	1.45
Sodium chloride	0.25	0.25	0.25
Zeolite	0.06	0.23	0.53
Vitamin-mineral premix ^†^	1.00 ^1^	1.00 ^2^	1.00 ^3^
Nutritional level			
ME (Mcal/kg) ^§^	2.90	3.00	3.10
CP, % ^‡^	20.98	18.22	16.38
Calcium, % ^‡^	1.06	0.87	0.83
Total phosphorus, % ^‡^	0.68	0.65	0.61
Lys, % ^§^	1.05	0.98	0.85
Met, % ^§^	0.46	0.40	0.34
Fe, mg/kg ^§^	50	50	50
Fe, mg/kg ^¶^	48.3	49.1	48.7

^†^ The vitamins and minerals (except Fe) in the basal diets were provided according to Chinese Feeding Standard of Chicken (2004). ^1^ Supplied per kilogram of starter diet: riboflavin, 9.0 mg; niacin, 60 mg; pantothenic acid, 16 mg; 50% cholinechloride, 800 mg; cobalamin,30μg; vitamin D_3_, 3300 IU; vitamin E (DL-α-tocophery acetate), 0.02 g; vitamin A (trans-retinyl acetate), 15,000 IU; vitamin K_3_, 6 mg; biotin,0.06 mg; folic acid, 1.5 mg; MnO, 100 mg; CuSO_4_•5H_2_O, 20 mg; ZnSO_4_•H_2_O, 150 mg; NaSeO_3_, 0.15 mg; KI, 0.5 mg. ^2^ Supplied per kilogram of grower diet: riboflavin, 8.0 mg; niacin, 48 mg; pantothenic acid, 16 mg; 50% cholinechloride, 1000 mg; cobalamin,15 μg; vitamin D_3_, 2750 IU; vitamin E (DL-α-tocophery acetate), 0.02 g; vitamin A (trans-retinyl acetate), 12,500 IU; vitamin K_3_, 5 mg; biotin, 0.05 mg; folic acid, 1.25 mg; MnO, 100 mg; CuSO_4_•5H_2_O, 20 mg; ZnSO_4_•H_2_O, 150 mg; NaSeO_3_, 0.15 mg; KI, 0.5 mg. ^3^ Supplied per kilogram of finisher diet: riboflavin, 8.0 mg; niacin, 44 mg; pantothenic acid, 16 mg; 50% cholinechloride, 800 mg; cobalamin,15 μg; vitamin D_3_, 2750 IU; vitamin E (DL-α-tocophery acetate), 0.02 g; vitamin A (trans-retinyl acetate), 10,000 IU; vitamin K_3_, 5 mg; biotin, 0.1 mg; folic acid, 0.75 mg; MnO, 100 mg; CuSO_4_•5H_2_O, 20 mg; ZnSO_4_•H_2_O, 150 mg; NaSeO_3_, 0.15 mg; KI, 0.5 mg.^§^ Values were calculated from data provided by the Feed Database in China (2018).^‡^ Determined by analysis; each value based on triplicate determinations.^¶^ Fe was analyzed by atomic absorption spectrophotometry.

**Table 2 animals-10-00670-t002:** Effects of dietary Fe on growth performance of Chinese yellow broilers during three 21-d experimental periods.

Variables ^‡^	Dietary Fe level, mg/kg						SEM ^†^	*p*-Value
	50	70	90	110	130	150		
ADG, g								
1 to 21 d	15.85	15.80	15.99	16.03	15.90	15.92	0.13	0.774
22 to 42 d	40.58	37.86	39.22	37.57	37.65	38.33	2.46	0.957
43 to 63 d	48.43	48.90	50.16	49.62	49.88	50.66	0.95	0.905
ADFI, g								
1 to 21 d	29.06	29.44	31.12	31.17	31.75	30.95	0.78	0.956
22 to 42 d	83.79	84.65	85.01	84.07	84.24	84.24	1.43	0.676
43 to 63 d	120.35	120.24	122.21	119.28	120.98	117.29	2.33	0.832
FCR								
1 to 21 d	1.83	1.86	1.95	1.94	2.00	1.94	0.04	0.261
22 to 42 d	2.06	2.24	2.17	2.24	2.24	2.20	0.02	0.816
43 to 63 d	2.49	2.46	2.44	2.40	2.43	2.32	0.08	0.933

^†^ SEM, standard error of the mean; n = 6. ^‡^ ADG: average daily gain; ADFI: average daily feed intake; FCR: feed conversion rate.

**Table 3 animals-10-00670-t003:** Effects of dietary Fe on index of immune organ of Chinese yellow broilers during three 21-d experimental periods.

Variables ^‡^	Dietary Fe level, mg/kg						SEM ^†^	*p*-Value
	50	70	90	110	130	150		
Spleen, % of BW								
1 to 21 d	0.171	0.187	0.177	0.153	0.171	0.210	0.026	0.195
22 to 42 d	0.168	0.171	0.182	0.178	0.166	0.163	0.008	0.805
43 to 63 d	0.184	0.190	0.192	0.178	0.188	0.183	0.012	0.702
Thymus, % of BW								
1 to 21 d	0.300	0.304	0.342	0.323	0.296	0.300	0.020	0.456
22 to 42 d	0.300	0.304	0.342	0.323	0.296	0.300	0.020	0.456
43 to 63 d	0.367	0.393	0.382	0.375	0.405	0.362	0.052	0.198
Bursa of Fabricius, % of BW								
1 to 21 d	0.491	0.448	0.526	0.528	0.504	0.443	0.046	0.430
22 to 42 d	0.353	0.376	0.390	0.366	0.378	0.429	0.034	0.603
43 to 63 d	0.115	0.121	0.132	0.114	0.140	0.116	0.020	0.267

^†^ SEM, standard error of the mean; n = 6. ^‡^ BW: body weight.

**Table 4 animals-10-00670-t004:** Effects of dietary Fe on the hematocrit of Chinese yellow broilers during three 21-d experimental periods and Fe content of liver and kidney at 63 d.

Variables	Dietary Fe level, mg/kg						SEM ^†^	*p*-Value
	50	70	90	110	130	150		
Hematocrit, %								
1 to 21 d	29.3	29.6	29.5	30.6	29.4	30.2	8.55	0.915
22 to 42 d	30.8	31.5	32.2	32.4	32.4	31.5	6.23	0.552
43 to 63 d	30.9	31.0	30.9	30.1	31.2	30.0	7.11	0.939
Fe, (mol /g)								
Liver 43 to 63 d	8.52	9.54	10.06	10.01	10.38	11.37	0.41	0.516
Kidney 43 to 63 d	5.65	6.04	6.04	6.73	6.48	7.52	0.24	0.262

^†^ SEM, standard error of the mean; n = 6.

**Table 5 animals-10-00670-t005:** Effects of dietary Fe between 43 and 63 d on meat quality of Chinese yellow broilers.

Variables			Dietary Fe level, mg/kg						SEM ^†^	*p*-Value
			50	70	90	110	130	150		
Breast muscle, %			5.43	5.50	5.39	5.55	5.52	5.47	0.12	0.974
Leg muscle, %			6.43	7.00	6.63	7.06	6.66	6.51	0.15	0.139
Meat color of breast muscle	45 min	L*	59.70	59.39	58.80	60.99	59.27	59.29	0.57	0.257
		a*	15.17	15.85	15.08	14.65	15.42	15.47	0.45	0.714
		b*	13.27	13.31	16.56	14.85	13.71	13.32	0.93	0.349
	24 h	L*	61.23	61.19	60.57	61.42	60.62	60.81	1.19	0.984
		a*	12.32 ^a^	12.71 ^a^	12.93 ^ab^	13.37 ^ab^	13.32 ^ab^	14.56 ^b^	0.15	0.041
		b*	14.52	14.57	16.99	15.90	14.72	15.00	0.38	0.083
pH of breast muscle	45 min		5.62 ^b^	5.79 ^ab^	5.97 ^a^	5.84 ^ab^	5.87 ^ab^	5.75 ^b^	0.06	0.001
	24 h		5.72	5.83	5.89	5.85	5.82	5.85	0.04	0.477
pH of leg muscle	45 min		6.12	6.13	6.17	6.21	6.19	6.24	0.05	0.342
	24 h		6.28	6.27	6.17	6.28	6.32	6.27	0.06	0.544
Drip loss of breast muscle, %			4.50 ^b^	4.22 ^b^	3.77 ^b^	4.29 ^ab^	4.14 ^b^	5.84 ^a^	0.45	0.024
Shear force of breast muscle, N			388.46	381.06	385.50	356.61	382.06	373.28	14.48	0.861

^†^ SEM, standard error of the mean; n = 6; ^a,b^ Means with different uppercase superscripts within the same row differ (*p* < 0.05). L*: relative lightness, a*: relative redness, and b*: relative yellowness.

**Table 6 animals-10-00670-t006:** Dose response regressions for Chinese Yellow broilers fed diets with different Fe content.

Variable	Model	Regression Equation ^1^	Dietary Fe Response, mg/kg	*p*-Value	R ^2^
pH of breast muscle (45 min)	QP ^2^	y = −0.0009x^2^ + 0.0194x + 4.8897	107.8	0.064	0.831
Drip loss of breast muscle	QP ^2^	y = 0.0005x^2^ − 0.0891x + 7.8401	89.1	0.069	0.841

^1^ Regression equations obtained using the analyzed Fe in the trial diets (50, 70, 90, 110, 130 and 150 mg/kg). ^2^ QP: Quadratic polynomial; QP model: Y = α + β × X + γ × X^2^, where Y is the response variable, X is the dietary Fe, α is the intercept; β and γ are the linear and quadratic coefficients respectively. The response was obtained by – β / (2 × γ).

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
