# Peer review of "Effects of Dietary Iron Level on Growth Performance, Immune Organ Indices and Meat Quality in Chinese Yellow Broilers"

_animals, 2020, doi:10.3390/ani10040670_

Round 1

Reviewer 1 Report

To my view, the initial idea of the study is not novel but interesting.

I accept the manuscript in that form.

Author Response

Response to Reviewer 1 Comments

Point 1: To my view, the initial idea of the study is not novel but interesting.

I accept the manuscript in that form.

Response 1: Thank you for the compliments.

Reviewer 2 Report

The authors answered to all my requests, so the paper can be accepted from my side.

Author Response

Response to Reviewer 2 Comments

Point 1: The authors answered to all my requests, so the paper can be accepted from my side.

Response 1: Thank you.

Reviewer 3 Report

Dear authors,

Thanks for the efforts with improving the manuscript:

L33: I would rephrase it the following way for more clarity: "The calculated final dietary Fe concentrations in Starter, Grower and Finisher phases was...".

Table 1: Vitamin-mineral premix: What is the reason vitamin D and K are supplemented at the same level in Starter and Finisher phases/studies and lower in Grower phase/study?

General comment: use Fe or "iron" consistently across the whole manuscript.

Round 2

Reviewer 3 Report

Check format Figure 1 and 2

Author Response

This manuscript is a resubmission of an earlier submission. The following is a list of the peer review reports and author responses from that submission.

Round 1

Reviewer 1 Report

The manuscript report an study about the effects of dietary iron level on growth performance, immune organ indices and meat quality in Chinese Yellow broilers. The hypothesis, M&M, results, discussion and conclusion are clear. However in my opinion the topic is no scientific novelty.

Reviewer 2 Report

The paper provide innovative and interesting data on Iron supplementation management in Chinese Yellow Broilers.

The manuscript is fine, only minor corrections are required.

line 24: explain QP (even if you explain at line 156, also you defined differently QP in line 156 and line 205, please check)

line 34: this is only a personal comment and a curiosity: I work with chickens since 20 years and I never thought to collect the whole themes (except for a sample for histology), how did you collected it completely? It's something very hard.

introduction: you put many emphasized own the effects of Fe on oxidative status, and I fully agree, but you did not evaluated this aspect in your manuscript, please reduce the comments on this topic, because not consistent with the data presented in the manuscript.

can you provide more general information on the genotype you used?  (performance mean data, phenotype) this would help the readers outside of China.

line 89 I'm quite astonished about the photoperiod (24 h light) you used. Is that allowed in China? I presume yes because you obtained ethical approval of the experiment, can you clarify?

table 1: the sum of ingredients is non 100 (but 99.94; 99,77 and 99,47 for the 3 periods respectively)

line 156 QP in line 205 is defined quadratic polynomial, please check and make uniform

lines 213-217: remove or reduce, you have the following reference on birds that is better, at least look for other reference for birds

229 and 241 bursa of Fabricius 

lines 239-241: in which species?

line 241: what does mean AA? is it a chicken genotype? please explain

line 242: remove gland

lines 242-243: which Fe-Gly range was applied? please report it

Reviewer 3 Report

Dear Authors,

I have several comments regarding the manuscript you submitted for publication at Animals.

General comments:

There are no clear indications why Chinese Yellow Broilers may have different requirements than fast-growing birds, based on their mg/day requirements. There are inconclusive results that can be explained by the great variability in the present study. This can partially be explained by: Limited number of replicates. The SEM of most of the parameters show that the number of replicates per treatment was very limited. Check SEM values for FCR in Table 2. These are not realistic. Wellbeing of birds: Why was lighting on for 24h a day? This can cause animal discomfort. 2 birds/pen were sacrificed at the end of each experiment. How were those birds sacrificed? Abstract: it should be clearly stated that 3 separate studies were conducted. 2 birds/pen in experiment 1 and 2 (Starter and Grower phases) were sacrificed only for spleen, thymus and bursa weighing. How is this justified? What were the indications that these parameters were relevant? I have the feeling this is not ethically justified.

Other comments:

L24: QP abbreviating needs to be explained. L32: 3 basal diets- clarify 1 per phase. L38: 24 – h? L59: references are not “pure” recommendation tables like NRC or Chinese Feeding tables, rather publications on challenging studies, etc. Were diets analysed for Fe to confirm the right inclusion rate? If so, please show the results. Was water samples analysed to confirm low Fe level? If so, show lab results. Table 1: Soy protein concentrate—was that regular SBM (=48% CP)? Or a product with greater CP content? Please indicate type of product and its CP level. L106: show vitamin and mineral premix content for all nutrients added in the premix. L181: “viz.”? Figures 1 and 2: arrows are not at the correct place.
